# Phosphorylation of ERK-Dependent NF-κB Triggers NLRP3 Inflammasome Mediated by Vimentin in EV71-Infected Glioblastoma Cells

**DOI:** 10.3390/molecules27134190

**Published:** 2022-06-29

**Authors:** Zelong Gong, Xuefeng Gao, Qingqing Yang, Jingxian Lun, Hansen Xiao, Jiayu Zhong, Hong Cao

**Affiliations:** 1Guangdong Provincial Key Laboratory of Tropical Disease Research, Department of Microbiology, School of Public Health, Southern Medical University, Guangzhou 510515, China; gongzeloong@foxmail.com (Z.G.); xuefenggao12121@163.com (X.G.); 15975564264@163.com (Q.Y.); lunjingxian@gmail.com (J.L.); gzlong1992@126.com (H.X.); z1402461564@126.com (J.Z.); 2Center Laboratory, Guangzhou Women and Children’s Medical Center, Guangzhou Medical University, Guangzhou 510120, China

**Keywords:** enterovirus 71, phosphorylation of ERK1/2, NLRP3, vimentin, NF-κB, glioblastoma cells

## Abstract

Enterovirus 71 (EV71) is a dominant pathogenic agent that may cause severe central nervous system (CNS) diseases among infants and young children in the Asia-pacific. The inflammasome is closely implicated in EV71-induced CNS injuries through a series of signaling pathways. However, the activation pathway of NLRP3 inflammasome involved in EV71-mediated CNS injuries remains poorly defined. In the studies, EV71 infection, ERK1/2 phosphorylation, and activation of NLRP3 are abolished in glioblastoma cells with low vimentin expression by CRISPR/Cas9-mediated knockdown. PD098059, an inhibitor of p-ERK, remarkably blocks the vimentin-mediated ERK1/2 phosphorylation in EV71-infected cells. Nuclear translocation of NF-κB p65 is dependent on p-ERK in a time-dependent manner. Moreover, NLRP3 activation and caspase-1 production are limited in EV71-infected cells upon the caffeic acid phenethyl ester (CAPE) administration, an inhibitor of NF-κB, which contributes to the inflammasome regulation. In conclusion, these results suggest that EV71-mediated NLRP3 inflammasome could be activated via the VIM-ERK-NF-κB pathway, and the treatment of the dephosphorylation of ERK and NF-κB inhibitors is beneficial to host defense in EV71-infected CNS.

## 1. Introduction

Enterovirus 71 (EV71) is a main pathogenic contributor to the hand, foot and mouth disease (HFMD), and is also thought to be a neuro-invasive virus that is commonly isolated from infants and young children with severe central nervous system (CNS) diseases [1,2,3]. EV71 causes sporadic epidemics and outbreaks, predominantly occurring in the Asia-Pacific [4]. The reason for death in EV71 infection is brainstem encephalitis due to pathogens invading the central nervous system [5]. From 2008 to 2014, EV71 infected more than 13 million people worldwide and elicited more than 3000 deaths, including 2457 deaths in China [6,7]. However, available treatments for EV71 infection are limited, especially for severe cases. Therefore, it is urgent to reveal the correlations between virus and host immune defense in term of inflammatory regulation and response, which may provide a potential basis on EV71 treatment.

Recent studies have shown that the innate immune recognition system plays an essential role against EV71 virus infection in humans [8,9,10]. Host cells may induce inflammatory responses through specific receptor proteins and cell signaling pathways [11,12], which results in inflammasome activation, subsequently executing its immune function against the pathogen. Inflammasome precisely regulated under a series of cell-stimulating signals contains several complex proteins [13]. The inflammasome can modulate caspase-1-dependent pyroptosis under infectious and stress-related contexts. In addition, as the member of the inflammatory family, the nod-like receptor protein 3 (NLRP3) is one of the most pivotal inflammasome related to CNS pathogenesis [14,15]. NLRP3 is the key to the host defense response via various pathogen-associated molecular patterns (PAMPs) [11]. Following the activation of NLRP3 inflammasome, the caspase-1 precursor (pro-caspase-1) transformed to active caspase-1. Active Caspase-1 induces the cleavage of pro-IL-1β into active IL-1β, recruits and activates other cells’ chemokines, amplifies the inflammatory response, and ultimately causes cell apoptosis and tissue injuries [8,9,10]. Studies targeting the pathogenic relationship between EV71 and NLRP3 inflammasome found that EV71 can bind to human macrophages and peripheral blood endothelial cells, which trigger the release of inflammasome-regulated product, and promotes the proliferation of EV71 in the host cell [16].

How does EV71 elicit severe inflammasome-related injuries in human CNS? Perison and his colleagues have confirmed that vimentin-dependent MAP kinases are essential in sciatic nerve axoplasm upon nerve injuries [17]. Research has concluded that ERK1/2 kinases are required in EV71 efficient replication [18]. In addition, according to the study of Tung WH, signaling pathways including MAPKs, NF-κB, and AP-1 were indispensable in enterovirus 71 replication [19]. Recently, a study with a novel treatment protocol for EV71 used a small molecular targeted NF-κB pathway [20]. Our group has verified the effect of vimentin on activation of EV71-induced NLRP3 inflammasome in vitro. Here, we designed a series of cellular experiments aiming to explore the complete signaling path-way triggered by EV71.

We hypothesize that the VIM-ERK-NF-κB signaling pathway activation induces NLRP3 inflammasome formation in EV71-infected glioblastoma cells. In this study, we interrogate the role of vimentin controlling EV71-induced inflammasome generation in vimentin-knockdown (VIM-KD) and Vimentin-wildtype (VIM-WT) glioblastoma cell models (U251 cells) to disclose the holistic crosstalk between EV71 and host. Meanwhile, PD098059 (the inhibitors of ERK) and CAPE (the inhibitors of NF-κB) were introduced to investigate the functions of ERK and NF-κB signaling molecules during the EV71-induced inflammasome production, respectively. These results warrant further exploration into effective therapeutic approaches to treat severe EV71-induced CNS disease.

## 2. Results

### 2.1. Vimentin Plays an Important Role in EV71-Induced Phosphorylation of ERK

The MAPK/ERK signaling pathway is composed of a series of signal proteins from a receptor on the cell surface to the DNA in the nucleus [17]. In order to examine whether EV71 can induce the activation of ERK phosphorylation during infection, U251 cells infected with EV71 were harvested at two hours and twenty-four hours post-infection. After purified cytoplasm and nuclear protein, western blot analysis was performed to detect p-ERK1/2 in U251 cells. Meanwhile, the GAPDH in both fractions were detected as internal loading controls. Results showed that EV71 can induce phosphorylation of ERK1/2 after two hours and twenty-four hours post-infection (Figure 1a,b). After that, we explored whether vimentin mediates phosphorylation of ERK in U251 cells infected with EV71. Vimentin knockdown U251 cells (VIM-KD) and wild type U251 cells (VIM-WT) were harvested after two hours and twenty-four hours EV71 post-infection. After purifying, western blot analysis was performed to detect the level of phospho-ERK1/2 (Thr202/Tyr204). Results showed that, the levels of p-ERK1/2 in U251 cells (VIM-WT) increased dramatically when compared with U251 cells (VIM-KD) after two hours and twenty-four hours post-infection (Figure 1c,d). Taken together, results suggest that phosphorylation of ERK is regulated by vimentin in U251 cells infected with EV71.

### 2.2. Phosphorylation of ERK Triggers the Activation of NF-κB Signaling Pathway

NF-κB pathway is a well characterized signaling pathway of both chemokines and cytokines [21]. A variety of viruses can damage the host cells via NF-κB signaling pathway [22]. It is essential to examine whether EV71 trigger the activation of NF-κB signaling in glioblastoma U251 cell. U251 cells were infected with EV71 and the cells were harvested after two and twenty-four hours post-infection. After purified cytoplasm and nuclear protein, western blot analysis was performed to detect NF-κB (p65) and IKBα in U251 cells by using antibodies. Meanwhile, the β-actin in both fractions were detected as internal loading controls. Results showed that the levels of p-ERK1/2 in U251 cells (VIM-WT) increased dramatically when compared with U251 cells (VIM-KD) after two hours and twenty-four hours post-infection. In the results, EV71 could induce a time-dependent promotion on the level of P65 in the nuclear protein (Figure 2c,d). The levels of IKBα decreased in U251 cell with the increase in infective periods. The results indicated that EV71 can trigger the activation of NF-κB signaling pathway in glioblastoma cells.

To determine whether the EV71-mediated NF-κB signaling pathway in the U251 cell could be regulated by ERK, the U251 cells were incubated with PD098059 (one kind of ERK inhibitors) for 60 min before being infected with EV71. Cells were harvested at two hours and twenty-four hours post-infection. Western blot analysis was performed to detect NF-κB (p65) both in the ERK inhibitor (PD098059) and control groups. Result showed that the levels of p65 increased in U251 cells with PD098059 (an ERK inhibitor) when compared to the control without PD098059 after EV71 infection (Figure 3b, *p* < 0.01). Results indicate that phosphorylation of ERK can trigger the activation of the NF-κB signaling pathway in U251 cells infected with EV71 that could be blocked by adding an ERK inhibitor (PD098059).

### 2.3. Vimentin Mediates the EV71/or VP1-Induced NF-κB Activation That Causing Brain Injuries

Vimentin has been reported as an attachment receptor for EV71 in previous studies [23]. In order to further determine the role of vimentin in EV71 infection process in glioblastomacells, a vimentin knockdown U251 cells (VIM-KD) were constructed using a lentivirus that stably expressed the sgRNA specific to vimentin by the CRISPR/Cas9 technique. Then vimentin protein was quantitative detected in VIM-KD cells. Results showed that vimentin had degraded dramatically in U251 cells (KD-VIM), compared to U251 cells (KD-WT) (Figure 4a,b). By using the VIM-KD cell model, we aim to explore whether vimentin mediates the activation of NF-κB signaling in U251 cells infected with EV71. U251 cells (VIM-KD) and U251 cells (VIM-WT) infected with EV71 were harvested after two hours and twenty-four hours post-infection. After purified cytoplasm and nuclear protein, Western blot analysis was performed to detect NF-κB (p65) in U251 cells. Results showed that after two hours and twenty-four hours post-infection, the levels of P65 declined distinctly in VIM-KD cells after EV71 infection, compared to U251 cells (VIM-WT), (Figure 4a). Taken together, results suggest that vimentin mediates the EV71-induced NF-κB activation in U251 cells.

In addition, VP1 protein (Abnova, China) of EV71 was involved in the study to further determine activators of NF-KB in U251 cells. U251 cells (VIM-KD) and U251 cells (VIM-WT) treated with VP1 protein (0.1 μg/mL) for two or twenty-four hours were collected following detection. After purified cytoplasm and nuclear protein, Western blot analysis was performed to detect NF-κB (p65) and IKBα in U251 cells. Results showed that the levels of P65 increased dramatically in U251 cells (VIM-WT) after exposure to VP1 protein, compared to the U251 cells (VIM-KD) (Figure 4e). Meanwhile, the levels of the IKBα declined in U251 cells (VIM-WT) after exposure to VP1 protein, compared to the U251 cells (VIM-KD) (Figure 4c,d). Figure 5 also show this in the KD-VIM cell. These results indicated that the activation of EV71-induced NF-κB signaling pathway can be mediated by vimentin, which can also be triggered by VP1 protein in U251 cells.

Moreover, to determine whether vimentin is important in inducing the EV-71 infection and damage of host cell directly, we evaluated EV71 replications and cytopathic effect (CPE) between U251 cells (VIM-KD) and U251 cells (VIM-WT) infected with EV71 (Figure 6a,c). The cells were infected with virus for three days at a concentration of 1 × 10^8^ TCID50 and MOI = 5. Under the inverted microscope, morphological profiles of cells were collected for analysis. In the results, (VIM-KD) U251 cells showed no significant changes in term of morphological profiles, while VIM-WT cells were observed with cell lesions, rounded profile, cell shrunken, shed and suspended in culture medium (Figure 6c). Such CPE changes were consistent with cellular EV71 replications in WT/KD cells. Our findings confirmed that vimentin plays a critical role in EV71-induced cytopathogenic injuries.

### 2.4. EV71-Induced Inflammasome Is Implicated in Vimentin/NF-κB Mediated CNS Injuries

It has been previously reported that EV-71 is a potent inducer of inflammation and NLRP3 inflammasome [24,25]. To further determine whether vimentin is required in EV71-induced inflammatory response, U251 cells (VIM-KD) and U251 cells (VIM-WT) infected with EV71 were harvested for six or twelve hours and then the levels of NLRP3 and caspase-1 were detected by Western blot in WT/KD cells. According to the results, after EV71 infection, the levels of both mature caspase-1 and NLRP3 inflammasome in U251 cells (VIM-WT) increased dramatically, while the equivalent indicators in U251 cells (VIM-KD) have no significant change (Figure 5a–c). In addition, quantitative real-time PCR assay of EV71 replication was applied in WT/KD cells at different times after infection, results confirmed that both NLRP3/Caspase-1 and cellular EV71 replications increased rapidly with the increment of exposure time.

Some previous studies validated that vimentin-mediated NF-κB signaling plays an important role in central nervous system infection [23,24]. Therefore, we sought to explore the role of vimentin-mediated NF-κB signaling pathway in the activation of inflammasome. CAPE is a specific inhibitor of NF-κB, which was involved in the EV71 infection experiment in vitro. In the results, the expression of caspase-1 and NLRP3 inflammasome declined remarkably in U251 cells (VIM-KD) and U251 cells (VIM-WT) with CAPE inhibitor, comparing with U251 cells (VIM-WT) (Figure 5d–f). These results suggest that vimentin and NF-κB are implicated in EV71-induced inflammasome activation in U251 cells.

### 2.5. A Proposed Schematic of EV71-Mediated NLRP3 Inflammasome Activation via VIM-ERK-NF-κB Pathway

Enterovirus 71 is a neurotropic virus, which may lead to severe neurological complications in children [26]. During the process of EV71 infection in the U251 cell, several signaling pathways could be triggered and finally result in cell injuries. There is a proposed schematic of EV71-mediated NLRP3 inflammasome activation via VIM-ERK-NF-κB pathway in U251 cells (Figure 7). In the schematic, EV71 and its virus proteins can get in touch with vimentin, which are located on the surface of the U251 cell firstly. Then, vimentin may activate EV71-induced phosphorylation of ERK, afterwards effecting the downstream NF-κB signaling pathway. Finally, NRLP3 inflammasome was induced via the VIM-ERK-NF-κB signaling pathway.

## 3. Discussion

HFMD is a self-limiting illness with typical symptoms, including mild fever and popular-vesicular rash, while neurological complications of HFMD may endanger patients’ life [27]. Severe CNS disease induced by EV71 is one of the most dangerous diseases among infants and young children, attributed to the rapid onset, high mortality, and serious sequelae [28,29]. Unfortunately, few efficient vaccines or anti-viral drugs are available in current clinical treatment [27]. Therefore, it is meaningful to disengage the exact pathological between EV71 and the host. Here, we show that ERK inhibitor and NF-κB inhibitor block the integrity of the VIM-ERK-NF-κB signaling pathway, which sequentially alleviate virus-induced injuries in the CNS.

Lines of studies have shown that activation of NRLP3 inflammasome is essential for host cells against EV71-induced CNS injuries [30,31]. NLRP3 inflammasome activation is a feedback loop event, which could subsequently result in a series of downstream cytokine production, while facilitating mature inflammasome formation [32]. Therefore, it should be prudent to unravel the relationships between NLRP3 inflammasome and EV71-induced CNS injuries [16]. It is putative that EV71 and its virus proteins are in touch with vimentin protein located on the surface of brain glioblastoma cells during the infection process, as seen in the proposed schematic paradigm [23,33]. Furthermore, vimentin promotes ERK phosphorylation, leading to NF-κB downstream event and NRLP3 inflammasome activation in glioblastoma cells upon EV71 insult.

Our results are reminiscent of several recent reports that kinds of pathogens take advantage of vimentin protein as an attachment receptor to host cells, ultimately contributing to the activation of NLRP3 inflammasome [34,35]. However, detailed mechanisms involved in vimentin-mediated inflammasome generation under EV-71 infection remain ambiguous. We pinpoint the role of vimentin in this study using VIM-WT and VIM-KD cell models. In addition, we utilize the PD098059 and CAPE to demonstrate the ERK-NF-κB signaling pathway upon EV71-induced inflammasome formation.

A compelling study has found that EV71 replication is highly related to MAPKs and NF-κB activation [19]. Later, another article proved that ERK1/2 is required for EV71 efficient replication in the infective cell [18]. At present, research is beginning to focus on the MEK/ERK signaling pathway in the EV71 infection. Evidence confirmed that berberine inhibits EV71 infection via regulating the MEK/ERK signaling pathway [36]. Strikingly, a subsequent study suggested that ERK regulates the function of the NF-κB signaling pathway by adjusting the activity of IKK in endothelial cells [37]. Based on our experiment results, it is reasonable to speculate that ERK regulates IKK phosphorylation, which contributes to the NF-κB nucleus translocation in EV71 infection.

It is urgent to seek efficient and available clinical treatments, as EV71 has already caused severe CNS disease among infants and children [16], Recently, several candidate vaccines showed protective effects against EV71 infection in neonatal mice [28]. Nevertheless, the vaccine strategy is hard to apply widely in the Asia-Pacific due to the prices. In this study, we confirmed that PD098059 and CAPE attenuates NRLP3 inflammasome activation via blockage of VIM-ERK-NF-κB singling cascades. Meanwhile, our results showed that VIM^−/−^ mice had limited CNS damage to WT mice in terms of the NLRP3 inflammasome, Caspase-1, IL-1β, and neurons in brain tissues (data are not shown here). Subsequently, we will draw attention to the effects of PD098059 and CAPE in the VIM^−/−^ mice model. Furthermore, nitric oxide (NO), an indicator of NF-κB mediated oxidative inflammatory responses, contributes to the development of inflammatory disease. A study suggested that inhibition of NF-κB downregulates the expression of inducible nitric oxide synthase (iNOS), resulting in a low level of NO production [38]. Our study will explore the inhibitory effect of PD098059 and CAPE on NO production following EV71 infection.

In summary, our current studies showed that EV71 takes advantage of vimentin, a protein located on the surface of U251 cells, to activate the downstream ERK phosphorylation and NF-κB translocation, ultimately facilitating NRLP3 inflammasome generation. Vimentin knockdown, phosphorylated ERK inhibitor, and NF-κB inhibitor could remarkably block these processes.

## 4. Materials and Methods

### 4.1. Cell Culture

The human glioblastoma (U251) cell lines, African green monkey kidney epithelial (Vero) cells were grown in monolayers at 37 °C in 5% CO_2_ in Dulbecco’s modified Eagle’s medium (DMEM, Gibco, Waltham, MA, USA), complemented with 10% inactivated fetal calf serum (FCS, PAN, Aidenbach, Germany), penicillin-streptomycin (Gibco, Waltham, MA, USA). During the whole process of cell experiments, cells were kept in DMEM unless otherwise indicated.

### 4.2. Virus and VP1 Protein

EV71 clinical strain involved in the study were collected from a 3-year-old child with severe hand-foot-mouth disease in Guangzhou Women and Children Hospital Central Laboratory. Three positive throat swab preservation solutions from the patient were filtered through a 0.45 μm filter, and 200 μL supernatants were inoculated into 6-well plates covered with monolayer of human RD cells in a carbon dioxide incubator at 37 °C. Duplicate test wells and negative control wells were also set in the meantime. After the appearance of a characteristic enterovirus cytopathic effect (CPE), the first generation of cultured cells and their medium were subjected to repeated cryopreservation for three times for lysis. Then the culture supernatant was collected and detected by q-PCR for identifying EV71. In addition, VP1 protein of EV71 was purchased from Abnova company in Hong Kong, China.

### 4.3. Construction of Vimentin Gene Knockdown Cell Line

To explore the role of vimentin and inflammasome in EV-71 infection of the CNS, we constructed a vimentin knockdown human U251 cell line by CRISPR/Cas9 technology. The four Vimentin gene Oligo single chain were designed and synthesized by Shanghai Yingjun company in China (Sequences as shown in Table 1). Linking the sgRNA fragment to lenti-CRISPRv2 plasmid was verified by sequencing and packaged with the lentivirus. U251 cells were transfected with the packaged and purified lentivirus. Finally, stable strains were selected through a screening procedure to obtain the gene knockdown U251 cell line.

### 4.4. CPE of U251 Cells Infected with EV71

The wild-type (VIM-WT) U251 cells and Vimentin gene knockdown (VIM-KD) U251 cells were inoculated into 7 cm cell culture dishes. After the cells were grown in full, the supernatant was removed. Cells were washed by PBS 3 times. Then fresh DMEM basal medium was added with EV71 at MOI = 5. Cytopathic changes were monitored by using the inverted microscope per day.

### 4.5. Immunoblotting Analysis

To determine whether EV71 or VP1 protein induces NF-κB activation in glioblastoma cells, U251 cell monolayers were grown on 60-mm plates. Confluent U251 monolayers were incubated with EV71 or VP1 protein (0.1 mg/mL) for 24 h. For the blocking assays of ERK with inhibitors, U251 monolayer cells were pre-incubated with or without PD098059 (KeyGEN, Nanjing, China) for 60 min, and then infected with EV71. After incubating, nuclear proteins and cytoplasmic of U251 cells were extracted with lysis buffer supplied with 1 mM Na3VO4 and 100 mM okadaic acid as described previously [20]. Both nuclear proteins and cytoplasmic were mixed with SDS buffer, heated and subjected to sodium dodecyl sulfate Polyacrylamide gel electrophoresis (SDS–PAGE). Samples were separated in 12% SDS-polyacrylamide gels and electroblotted onto polyvinylidene difluoride membranes (Millipore, Burlington, MA, USA). After blocking with blocking reagent (including 5% milk in PBS with 0.1% Tween 20) overnight, cytoplasm proteins from cells membranes were probed with antibodies against phospho-ERK1/2 (Thr202/Tyr204) (0.2 mg/mL), IkBa (0.2 mg/mL, Cell Signaling Technology, Danvers, MA, USA), GAPDH (0.1 mg/mL) for 2 h, respectively. The nuclear proteins from membranes were probed with antibodies against NF-kB (p65) (0.4 mg/mL), and GAPDH (0.1 mg/mL) for 2 h, respectively. The harvested membranes were incubated with a horseradish peroxidase (HRP)-conjugated secondary antibody for 1 h at room temperature, and then visualized using an enhanced chemiluminescence procedure (Roche Applied Science, Indianapolis, IN, USA).

### 4.6. Statistical Analysis

Measurement data involved in the study are presented as mean ± SD. The data was analyzed by using SPSS 19.0 software. The variances between groups were analyzed by one-way ANOVA (SNK-q test was used to compare the two groups). Other experiments were analyzed by Student’s *t* test. *p* < 0.05 was considered to be of statistical significance.

## 5. Conclusions

In conclusion, these results suggest that EV71-mediated NLRP3 inflammasome could be activated via VIM-ERK-NF-κB pathway, and the treatment of the dephosphorylation of ERK and NF-κB inhibitors are beneficial to host defense in EV71-infected CNS.

## Figures and Tables

**Figure 1 molecules-27-04190-f001:**
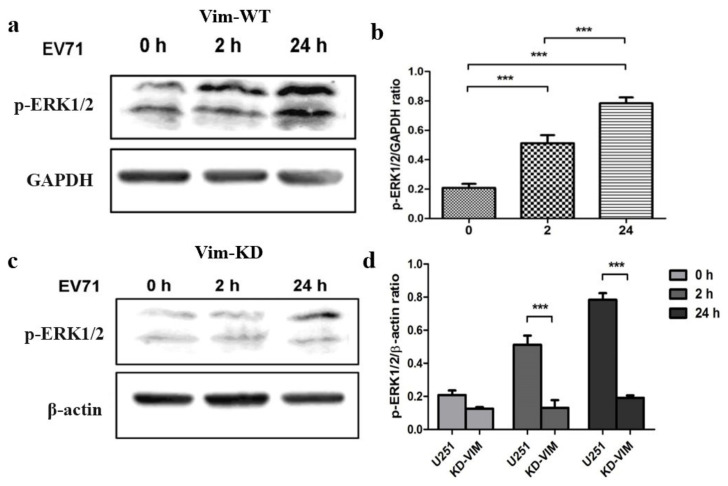
EV71 infection induced activation of NF-κB signaling. U251 cell was infected with EV71 for two and twenty-four hours, (**a**) Western immunoblotting analysis of ERK phosphorylation expression level at different times after infection. (**b**) p-ERK and GAPDH ratio by gray analysis of Western immunoblotting. (**c**,**d**) Inhibition of phosphorylation of ERK by knockdown of vimentin. (**a**,**c**) Vimentin knockdown U251 cells (KD-VIM) and wild type U251 cells (U251) were infected with EV71 for two and twenty-four hours. Western immunoblotting analysis ERK phosphorylation expression level. (**d**) p-ERK and β-actin ratio by gray analysis of Western immunoblotting. *** *p* < 0.0001.

**Figure 2 molecules-27-04190-f002:**
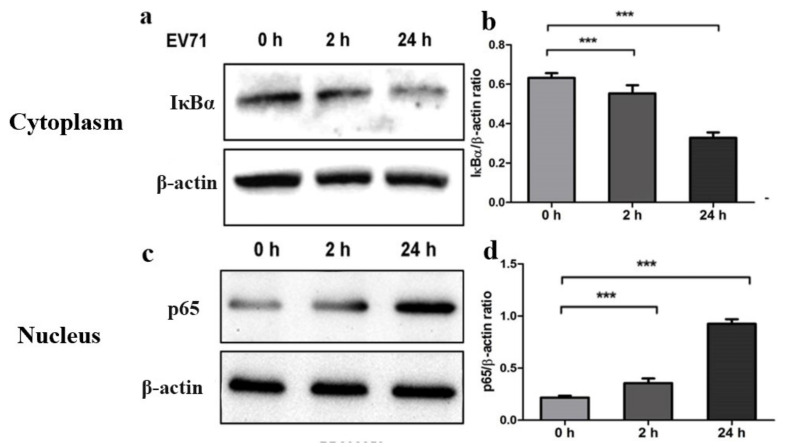
EV71 induced the activation of NF-κB in U251 cell. (**a**,**c**) NF-κB (p65) and IKBα were examined after EV71 infection, comparing with P65 and IKBα expressions at zero hours, two hours and twenty-four hours, respectively. IKBα and NF-κB (P65) expression level detected by Western immunoblotting analysis. After infection, P65 was up-regulated dramatically in the nucleus, while the level of IKBα decreased within twenty-four hours (*p* < 0.01). (**b**,**d**) IKBα/NF-κB (p65) and β-actin ratio analysed by gray analysis of Western immunoblotting. *** *p* < 0.0001.

**Figure 3 molecules-27-04190-f003:**
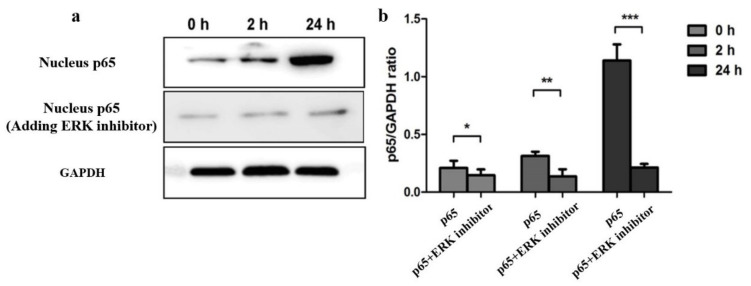
Phosphorylation of ERK mediates the activation of NF-κB signaling pathway in enterovirus 71 infection. (**a**) Inhibition of phosphorylation of ERK declined the level of NF-κB (p65) in the nucleus after EV71 infection. NF-κB (P65) expression level both in Control and ERK inhibitor (PD098059) groups by using Western immunoblotting analysis. (**b**) NF-κB (P65) and GAPDH ratio analyzed between the ERK inhibitor (PD098059) and control groups by gray analysis of Western immunoblotting. * *p* < 0.05, ** *p* < 0.01 and *** *p* < 0.0001.

**Figure 4 molecules-27-04190-f004:**
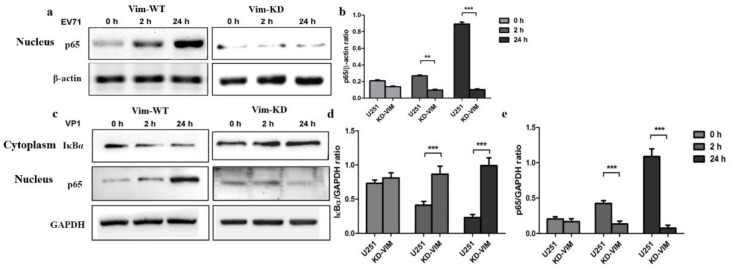
Role of vimentin in EV71 (VP1)-induced NF-κB activation. (**a**,**b**) Role of vimentin in EV71-induced NF-κB activation in nucleus. The levels of NF-κB (p65) translocation to the nucleus were examined in nuclear fractions at zero hours, two hours and twenty-four hours. (**a**) The levels of NF-κB (P65) were detected by Western immunoblotting. (**b**) NF-κB (P65) and β-actin ratio analyzed by gray analysis of Western immunoblotting. (**c**–**e**) Role of vimentin in VP1-induced NF-κB activation. (**c**) The levels of IKBα and NF-κB (P65) detected by Western immunoblotting analysis, respectively. (**d**,**e**) IKBα or NF-κB (P65) and GAPDH ratio analyzed by gray analysis of Western immunoblotting, respectively. ** *p* < 0.01, *** *p* < 0.0001.

**Figure 5 molecules-27-04190-f005:**
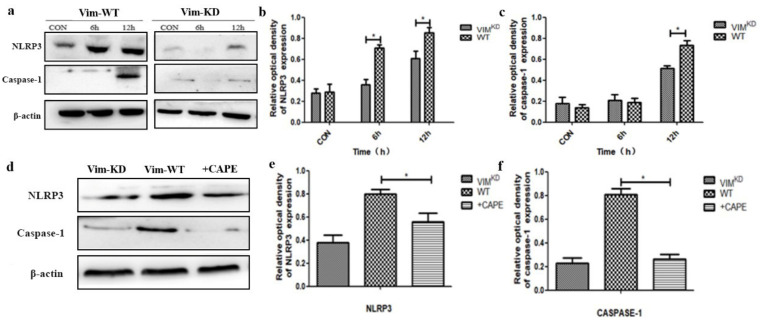
Vimentin and NF-κB signaling pathway are essential for the EV-71-induced inflammasome activation. (**a**) The expressions of NLRP3 and caspase-1 protein detected by Western blotting between VIM^+/+^ cells and VIM^−/−^ cells. (**b**,**c**) Optical density analysis of NLRP3 and Caspase-1 by Western blotting. (**d**) Western blot detected the NLRP3 and caspase-1 protein expression among VIM^+/+^ cells, VIM^−/−^ cells and CAPE cells. (**e**,**f**) NLRP3 or Caspase-1 and β-actin ratio analyzed by gray analysis of Western immunoblotting. * *p* < 0.05.

**Figure 6 molecules-27-04190-f006:**
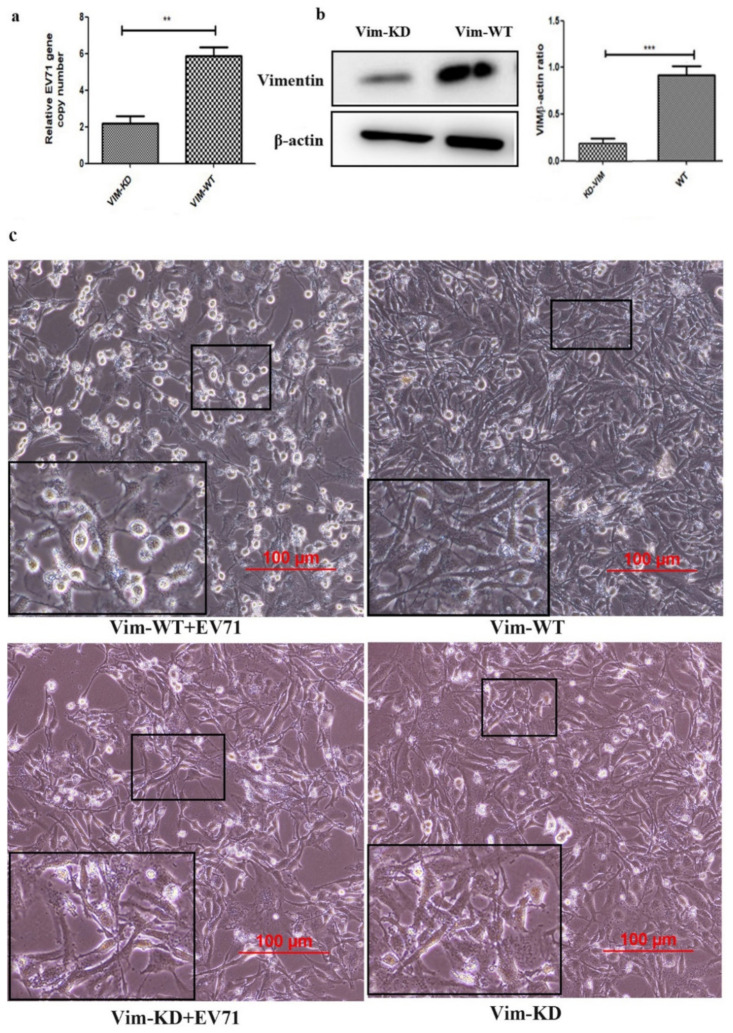
Vimentin mediates the CPE of EV-71 infection. (**a**) Quantitative real-time PCR assay of EV71 replication in VIM-WT and VIM-KD U251 cells. (**b**) The expression of vimentin in VIM-WT U251cells is much higher than that in VIM-KD U251 cells, analyzed by gray analysis of Western immunoblotting. (**c**) Morphological profiles of U251 cells (left: VIM^+/+^ cells infected with EV-71; middle: VIM^−/−^ cells infected with EV-71; right: uninfected U251 cells). ** *p* < 0.01, *** *p* <0.0001.

**Figure 7 molecules-27-04190-f007:**
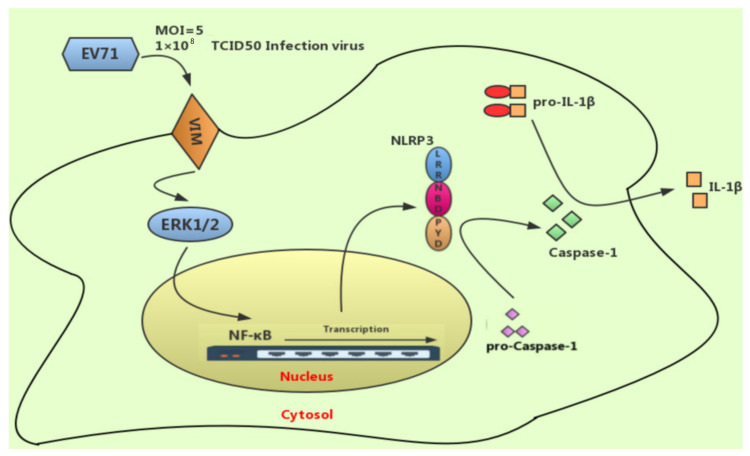
A proposed schematic of EV71-mediated NLRP3 inflammasome activation via VIM-ERK-NF-κB pathway in U251 cells.

**Table 1 molecules-27-04190-t001:** The designed sgRNA oligonucleotide strands.

Vim1 F	5-GGAGCGCGACAACCTGGCCG NGG-3
Vim1 R	5-CGGCCAGGTTGTCGCGCTCC NGG-3
Vim2 F	5-GCGCACGGCAGAGGAGCGCG NGG-3
Vim2 R	5-CGCGCTCCTCTGCCGTGCGC NGG-3

## Data Availability

Not applicable.

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
