# Peer review of "Phosphorylation of ERK-Dependent NF-κB Triggers NLRP3 Inflammasome Mediated by Vimentin in EV71-Infected Glioblastoma Cells"

_molecules, 2022, doi:10.3390/molecules27134190_

Round 1
Reviewer 1 Report
Revision of the paper:
“Phosphorylation of ERK-dependent NF-κB triggers NLRP3 inflammasome mediated by vimentin in EV71-infected astrocyte cells” by Ze-Long Gong, Xue-Feng Gao, Jia-Yu Zhong, Han-Sen Xiao, Qing-Qing Yang, Hong Cao
In the present paper the authors explored signalling pathways after EV71 entry into astrocytes. In the present paper authors show that EV71-mediated NLRP3 inflammasome could be activated via VIM - ERK - NF-κB pathway, and the treatment of the dephosphorylation of ERK and NF-κB inhibitors are beneficial to host defense in EV71-infected CNS.
The paper would be of interest to readers, since the activation pathway of NLRP3 inflammasome involving in EV71-mediated CNS injuries is poorly defined. Please find the following suggestions to improve the text:
Title: instead astrocyte, it would be more proper to you use glioblastoma, as U251 are glioblastoma cell line. Primary astrocytes may react differently to infection.
Abstract line 17; astrocytes lacking vimentin: please correct by changing actrocytes into U251, and lacking vimentin is not the correct term, since Figure 5 still shows some vimenitn in WB.
Introduction section: For a broader audience, readers would benefit, if also a virus family will be quoted. It is important to cite appropriate references of how many cases of EV17 CNS entry have been reported globally.
The introduction section should be modified in a way that it will read more fluently. Please avoid the style “this researchers showed this…. , this researches showed that…. “.
The Material and Methods section: Why did you use MOI 5? It is rather high.
Line 271: ….vimentin protein which located on the surface of brain astrocytes cells… Please include reference, the statement is rather unconfirmed. It is localized in the cytoplasm.
Figure 5: It appears that you have some surplus words in a) groups, strains… ? CPE is poorly visible, lease show better image. CON : you have to show controls for each of uninfected cells. Vim +/+ are also U251, as are Vim -/-.
Figure 7: Cell surface localization of vimentin needs to be verified. Or at least references added.
How the expression levels of vimentin alter after EV71 infection?
Author Response
Dear reviewer,
It is our great pleasure to receive your decision and advice on our manuscript “Phosphorylation of ERK-dependent NF-κB triggers NLRP3 in-flammasome mediated by vimentin in EV71-infected astrocyte cells” (Manuscript ID: molecules-1764978). We thank you and the reviewers for the review of our manuscript.
We have addressed the comments raised by the reviewers, and the amendments are highlighted in red in the revised manuscript.The coverletter responding to the review comments were uploaded to the systems, please see the attachment. Since only one file can be uploaded here, we place the content of the cover letter at the last (L561-707) of the revised manuscript as a file upload system.
We wished that the quality of our manuscript can meet Moleculars requirements. It is our honor to have the opportunity to publish this manuscript on Moleculars.
Best wishes.

Reviewer 2 Report
Journal: Molecules
Manuscript ID: molecules-1764978
Type of manuscript: Article
Title: Phosphorylation of ERK-dependent NF-κB triggers NLRP3 inflammasome
mediated by vimentin in EV71-infected astrocyte cells
Authors: Zelong Gong, Xuefeng Gao, Jiayu Zhong, Hansen Xiao, Yang Qingqing,
Hong Cao *
Submitted to section: Medicinal Chemistry,
As mentioned in conclusion, this paper described EV71-mediated NLRP3 inflammasome could be activated via VIM-ERK-NF-κB pathway tracking mainly by cell culture and Western blotting. In particular, the authors revealed the role of vimentin in the sequence of events in vitro.
The experiment is simple and the description is concise and enough. The previously known findings are well described. In particular, the novel aspects and limitation of the paper are clearly presented in L275-287.
Therefore, This reviewer recommends that this reviewer publish this paper in Molecules as an article.
Comment and question
1. As the authors may well known, most of the factors in the current paper are incorporated in very complex pathways. So, from a medicinal point of view, side effects or off-target are a concern, is it possible to discuss this issue? And more, this reviewer thinks this article is not match to the journal, especially, the section of Medicinal Chemistry. ( https://www.mdpi.com/journal/molecules/sections/medicinal_chemistry )
2. Depending on the cell and the experiment, β-actin or GAPDH are used as the loading control. Is it possible to unify one or the other? Otherwise, please describe the justification for using both. Also, in the figures, the font of protein names are not unified.
3. L317, L320
NO, nitric oside (NO): change the order
Author Response
Dear reviewer,
It is our great pleasure to receive your decision and advice on our manuscript “Phosphorylation of ERK-dependent NF-κB triggers NLRP3 in-flammasome mediated by vimentin in EV71-infected astrocyte cells” (Manuscript ID: molecules-1764978). We thank you and the reviewers for the review of our manuscript.
We have addressed the comments raised by the reviewers, and the amendments are highlighted in red in the revised manuscript.
The coverletter responding to the review comments were uploaded to the systems, please see the attachment. Since only one file can be uploaded here, we place the content of the cover letter at the last (L561-600) of the revised manuscript as a file upload system.
We wished that the quality of our manuscript can meet Moleculars requirements. It is our honor to have the opportunity to publish this manuscript on Moleculars.
Best wishes.

Round 2
Reviewer 2 Report
Journal: Molecules
Manuscript ID: molecules-1764978 _(R1)
Type of manuscript: Article
Title: Phosphorylation of ERK-dependent NF-κB triggers NLRP3 inflammasome
mediated by vimentin in EV71-infected glioblastoma cells
Authors: Zelong Gong, Xuefeng Gao, Yang Qingqing, Jingxian Lun, Hansen Xiao, Jiayu Zhong,
Hong Cao *
Submitted to section: Medicinal Chemistry,
As mentioned in conclusion, this paper described EV71-mediated NLRP3 inflammasome could be activated via VIM-ERK-NF-κB pathway tracking mainly by cell culture and Western blotting. In particular, the authors revealed the role of vimentin in the sequence of events in vitro.
The experiment is simple and the description is concise and enough. The previously known findings are well described. In particular, the novel aspects and limitation of the paper are clearly presented.
Therefore, This reviewer recommends that this reviewer publish this paper in Molecules as an article.
Comment and question
It is the understanding of this reviewer that the paper is concise and adequate. Although the paper has undergone significant revisions, again, the reviewer did not find an argument from a medicinal chemistry standpoint, a point that this reviewer pointed out last time.
The final decision will be made by the editor, but the reviewer recommends that the manuscript be submitted to another section or another journal.the section of Medicinal Chemistry. ( https://www.mdpi.com/journal/molecules/sections/medicinal_chemistry )